# Prediction Algorithm for Satellite Instantaneous Attitude and Image Pixel Offset Based on Synchronous Clocks

Lingfeng Huang [1,2,3], Feng Dong [1,2] and Yutian Fu [1,2,*]

1 Shanghai Institute of Technical Physics, Chinese Academy of Sciences, Shanghai 200083, China
2 Key Laboratory of Infrared System Detection and Imaging Technology, Chinese Academy of Sciences, Shanghai 200083, China
3 University of Chinese Academy of Sciences, Beijing 100049, China
* Correspondence: yutianfu@mail.sitp.ac.cn

**Abstract:** To ensure a high signal-to-noise ratio and high image volume, a geostationary orbiting ocean remote-sensing system needs to maintain high platform stability over a long integration time because it is affected by satellite attitude changes. When the observation target is the ocean, it is difficult to extract image features because of the lack of characteristic objects in the target area. In this paper, we attempt to avoid using image data for satellite attitude and image pixel offset estimation. We obtain the satellite attitude by using equipment such as gyroscopes and performing time registration between the satellite attitude and the image data to achieve pixel offset matching between images. According to the law of satellite attitude change, we designed a Kalman-like filter fitting (KLFF) algorithm based on the satellite attitude change model and the Nelder–Mead search principle. The discrete attitude data were time-marked by a synchronization system, and high-precision estimation of the satellite attitude was achieved after fitting with the KLFF algorithm. When the measurement accuracy of the equipment was $1.0 \times 10^{-3\circ}$, the average prediction error of the algorithm was $1.09 \times 10^{-3\circ}$, 21.58% better than the traditional interpolation prediction result of $1.39 \times 10^{-3\circ}$. The peak value of the fitting angle error reached $2.5 \times 10^{-3\circ}$. Compared with the interpolation prediction result of $6.2 \times 10^{-3\circ}$, the estimated stability of the satellite attitude improved by about 59.68%. After using the linear interpolation method to compensate for the estimated pixel offset, its discrete range was 0.697 pixels. Compared with the 1.476 pixels of the interpolation algorithm, it was 52.8% lower, which improved the noise immunity of the algorithm. Finally, a KLFF algorithm was designed based on the satellite attitude change model by using external measurement data and the synchronous clock as a benchmark. The instantaneous attitude of the satellite was accurately estimated in real time, and the offset matching between the images was realized, which lays the foundation for in-orbit satellite data processing.

**Keywords:** satellite attitude estimation; remote-sensing system measurement; adaptive fitting; time synchronization mark; featureless registration

## 1. Introduction

To obtain a better image signal-to-noise ratio from a geostationary orbiting ocean remote-sensing system, a sufficiently long integration time is required. Due to the stability limitations of the platform, the satellite imaging system has attitude deviation, which causes image distortion [1]. Usually, the long integral image is divided into time-division integral images on which a geometric distortion correction is performed by means of image feature matching [2,3]. Some studies use scale-invariant feature transform (SIFT), speeded-up robust features (SURF), or other extraction methods to register remote-sensing images [4–6]. At the same time, some studies have pointed out that a multi-sensor registration system can be used for image registration [7,8]. However, in ocean imaging [9], it is difficult to extract features due to the lack of conspicuous surface objects. Therefore, some researchers

have avoided using image information and used inertial navigation, gyroscopes, star sensors, and other equipment to measure the instantaneous attitude information of the platform [10,11] and used high-precision instantaneous attitude information to repair the geometric image pixel offset [12,13].

The pixel size of a marine remote-sensing satellite camera is 18 μm, the optical focal length is 3226 mm, and the angular resolution of the pixel is calculated as $3.19 \times 10^{-4\circ}$, for which high-precision attitude measurement is required. Some studies achieved attitude angle estimation with an accuracy of about 0.03° through an extended Kalman filter [14], but it consumed a lot of computing power. There has also been research into the attitude of high-orbit targets through space-based observation facilities [15] and using a new fiber-optic gyroscope design to improve the zero-bias stability and measurement accuracy [16]. Some research groups adopted the new optical design of a high-precision star camera and high-precision measurement method [17,18] and achieved an angle measurement accuracy of $1.5 \times 10^{-4\circ}$, but the output frequency of high-precision data was limited, generally in the range of 10–20 Hz. Some studies have pointed out that the frame frequency of complementary metal oxide semiconductor (CMOS) detectors in remote sensing can be as high as 500 Hz [19,20]. According to this, the maximum frame frequency of the visible light band is designed as 100 Hz, which is much bigger than the attitude sensor's 10–20 Hz frequency.

For the problem of frame rate mismatch between the satellite attitude and camera data, an interpolation algorithm is often used for attitude estimation. For example, the Beidou-3 satellite uses the interpolation method to estimate the image pixel offset by calculating the time interval with adjacent attitude data and applying different weights according to the interval [21]. Some research groups also use the Kalman filter method to fit the satellite tremor measurement data nonlinearly [22] and realize the tremor distortion recovery by constructing a tremor model. When the satellite is in orbit, the attitude angle changes according to a certain law so that the high-precision attitude modeling of the satellite can be carried out. For example, the ZY-3 satellite follows a sinusoidal law in the single-axis direction [23]. Mathis Bloßfeld et al. preprocessed data from the Jason series satellites, established a model for the satellite attitude, and supplemented the missing data through linear and spherical-linear interpolation. Through data preprocessing, in the precise orbit determination (POD), the root-mean-square (RMS) of the attitudes of Jason series satellites 1, 2, and 3 improved by 5.93, 8.27, and 4.51%, respectively, which improved attitude determination accuracy [24]. Sylvain Loyer et al. performed data exchange and fusion on the attitude data of multiple satellites, where satellite attitude was represented by quaternions. After correcting and registering the time and attitude quaternions between different satellites, the precise point positioning (PPP) RMS of the satellites was reduced by 50%, which indicated a more accurate attitude positioning result [25]. One study achieved attitude estimation through the fusion of data from sensors and satellite navigation systems [26], and [27] performed data fusion through multiple microsensors to confirm the attitude of nanosatellites.

When the characteristics of the observation target are not obvious, it is difficult to use feature extraction methods such as SURF to calculate the image pixel offset. At this time, external measurement equipment such as gyroscopes and star cameras are often used for auxiliary measurement. The data measured by external measurement equipment are usually high-precision, low-frequency data, which cannot correspond to high-frequency image data. If traditional interpolation algorithms are used for estimation, a large amount of measurement noise will be introduced. Considering the above limitations, to improve the estimation of satellite attitude and image pixel offset when there is no characteristic target, in this study we attempt to time stamp the high-precision, discrete attitude data through a time synchronization system and design a Kalman-like filter fitting (KLFF) algorithm based on the attitude change model of the imaging platform and the Nelder–Mead search principle. The algorithm can then fit the data to the attitude prediction function of the corresponding satellite attitude change model after denoise filtering and

perform continuous, real-time adaptive optimization through the latest measured attitude. The instantaneous satellite attitude data at any time are calculated through the marked time, and the transformation matrix of the optical imaging system is introduced to solve the estimated satellite attitude as the optical axis moves. Finally, the KLFF algorithm can obtain the estimated image pixel offset with the camera parameters. In contrast to existing traditional interpolation attitude estimation algorithms, we set out to improve the accuracy and noise immunity of attitude and pixel offset estimation. By constructing an attitude estimation function, the instantaneous attitude calculation of the satellite was decoupled from the attitude-measuring equipment, thereby improving the real-time performance of the satellite attitude and pixel offset estimation algorithm. For the estimated attitude, image pixel offset compensation was performed using centroid extraction [28] and linear interpolation. The result was compared with the interpolation algorithm compensation and the true centroid position to evaluate the effect of the algorithm.

## 2. Materials and Methods

### 2.1. Principle of Satellite Attitude Prediction Algorithm

In a geostationary orbiting ocean imaging system, after using multiple exposures and accumulations, the multi-spectrum and high signal-to-noise ratio require that the stability of the whole star image reaches $1.9 \times 10^{-5\circ}$/s, which is a high requirement. If satellite attitude measurement data are used to match the phase shift with data before the accumulation of multiple exposures, the stability requirements for satellite attitude can be greatly reduced. Some studies have pointed out that a global clock system can achieve high accuracy and low time drift and can be stable within 40–140 ns [29,30]. In addition, some re-search groups have carried out high-precision, high-stability time stamping on other data, such as images, by adding controllers and ciphers [31,32]. Based on these, we propose a high-precision, low-drift clock to generate the time stamp of image and satellite attitude data and then to perform discrete data fitting based on the time stamp and align the attitude and image data in real time. Referring to existing research, the attitude changes of geostationary satellites follow high- and low-frequency laws [33], and their vibrations consist of low-frequency periodic and high-frequency fine vibrations according to the sine law [34–36]. In [34], Alfred S. McEwen analyzed the scientific data of HiRISE (The High Resolution Imaging Science Experiment) on the MRO (Mars Reconnaissance Orbiter) and proposed that the registration data of the overlapping parts of the CCDs revealed the sinusoidal motion of the spacecraft along and across the orbits. In [35], P. Schwind studied the attitude change in the EnMAP (Environmental Mapping and Analysis Program). From the satellite's geometric simulator, the sinusoidal law model of satellite attitude change was proposed and constructed. In [36], Jun Pan analyzed the attitude jitter of the satellite, constructed a model of the satellite attitude change, and verified the model with the actual images of Ziyuan-3 and Ziyuan1-02C. They proposed a model for the composition of the sinusoidal signals of different frequencies and simplified the model by using a simple sinusoidal function to characterize the attitude change in the satellite. Therefore, based on the characteristics of the image and satellite attitude data, we constructed the corresponding experimental and the schematic diagrams as shown in Figure 1.

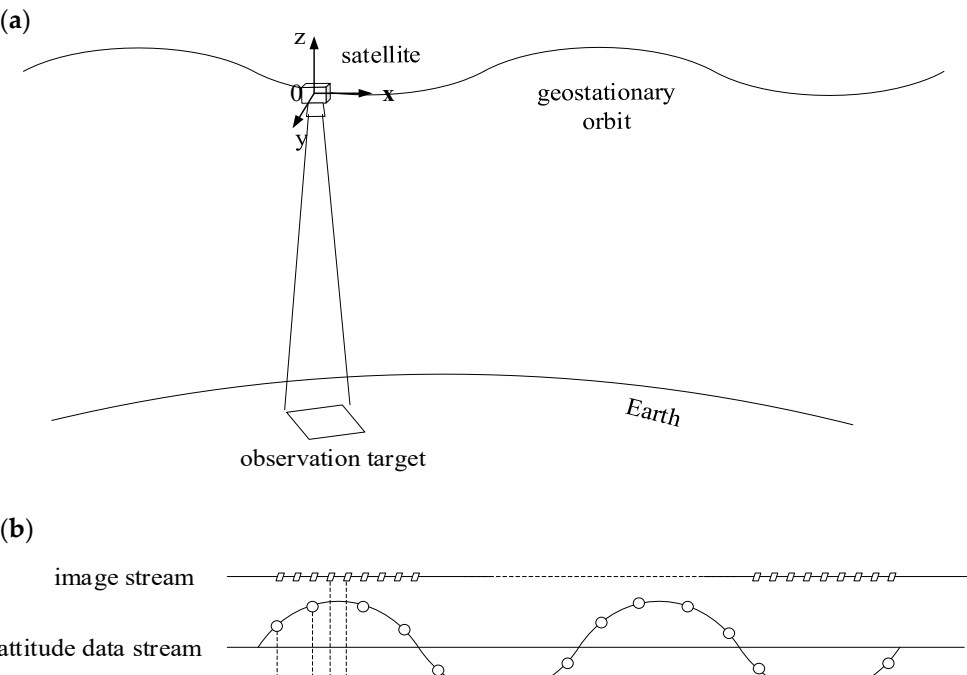

**Figure 1.** Satellite remote-sensing imaging system. (**a**) Schematic diagram of the geostationary orbit satellite imaging system; (**b**) schematic diagram of mismatch between discrete attitude data of sine law and discrete image data.

Referring to the principle of the Kalman filter [37,38], we propose a Kalman-like filter fitting (KLFF) algorithm based on the satellite attitude change model. The algorithm flow is shown in Figure 2. It is divided into three modules: initial parameter prefitting, adaptive real-time fitting, and image and attitude time registration. The algorithm uses attitude sensors such as gyroscopes and star cameras to collect satellite attitude data and uses a camera system connected to attitude sensors to collect remote-sensing images. The satellite attitude data and images are time-stamped by the synchronization clock (CPU clock in this paper). The parameter prefitting and the adaptive real-time fitting modules cooperate to complete the fitting of the attitude estimation function. The image and attitude time registration module substitutes the image's time stamp into the attitude prediction function, performs a geometric optics calculation according to the optical parameters, and finally obtains the image pixel offset.

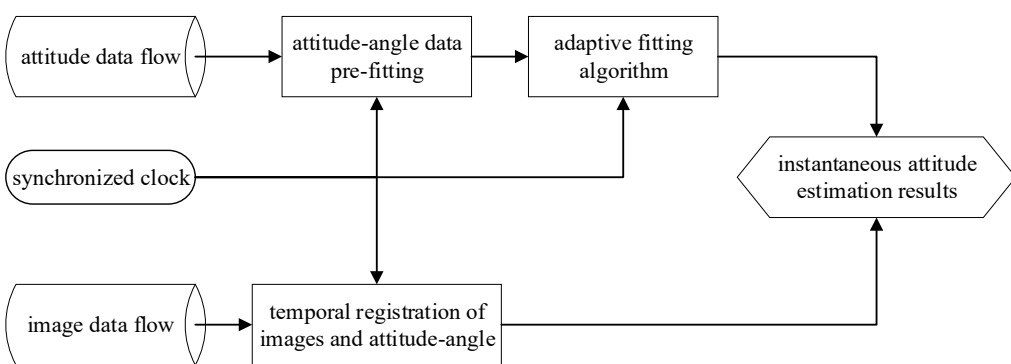

**Figure 2.** Flowchart of the lightweight adaptive algorithm for instantaneous image pixel offset solutions.

Based on a specific satellite attitude change model, such as the sinusoidal model used in this paper, the KLFF algorithm was designed, as shown in Equation (1). The attitude estimation function is $F$; the parameter optimization process is function $H$; and the Kalman-like filter process is $K$.

$$\begin{cases} X_{(n|n-1)} = F(\boldsymbol{P}_\mathrm{m}, t_\mathrm{n}) = \varepsilon + A_\mathrm{mp} \times \sin(\omega \times t_\mathrm{n} + \varphi) \\ X_{(n|n)} = K\left(X_{(n|n-1)}\right) \\ \boldsymbol{P}_{\mathrm{m}+1} = H\left(\boldsymbol{P}_\mathrm{m}, X_{(n|n)}\right) \quad \mathrm{m}, \mathrm{n} = 0, 1, 2 \cdots \\ \boldsymbol{P}_\mathrm{m} = \begin{bmatrix} R_\mathrm{es} & A_\mathrm{mp} & \omega & \varphi & \varepsilon \end{bmatrix} \end{cases} \tag{1}$$

In Equation (1), $F(\boldsymbol{P}_\mathrm{m}, t_\mathrm{n})$ is the attitude estimation function based on the satellite attitude change model; $\varepsilon$ is the attitude estimation error; and $A_\mathrm{mp}$, $\omega$, and $\varphi$ are the amplitude, frequency, and initial phase of the estimation function, respectively. $\begin{bmatrix} A_\mathrm{mp} & \omega & \varphi & \varepsilon \end{bmatrix}$ form a parameter array designated $\boldsymbol{P}_\mathrm{m}$, where m is the parameter serial number. $t_\mathrm{n}$ is the specific time instant at time stamp n marked by the synchronous clock, and n is the data sequence number. Since the parameters are not refitted at every time n, m and n are not exactly equal. By performing a Fourier transform on the attitude acquisition data with sinusoidal periodicity, the initial parameters $\boldsymbol{P}_0$ were obtained according to the position and amplitude of the main peak in the frequency spectrum. $X_{(n|n-1)}$ represents the estimated attitude data calculated using the attitude estimation function $F$ at time n + 1 when the parameter is $\boldsymbol{P}_\mathrm{m}$. $X_{(n|n)}$ represents the attitude data after the Kalman-like filtering of $X_{(n|n-1)}$. These data and the current function parameter $\boldsymbol{P}_\mathrm{m}$ were used as input data for the function $H$, and KLFF was performed to obtain the latest function parameter $\boldsymbol{P}_{\mathrm{m}+1}$. Through the above steps, the closed-loop calculations of the attitude estimation function parameters $\boldsymbol{P}_\mathrm{m}$, the estimated attitude $X_{(n|n-1)}$, the Kalman-like filtering data $X_{(n|n)}$, and the new attitude estimation function parameters $\boldsymbol{P}_{\mathrm{m}+1}$ were realized, and the parameter adaptive optimization based on the attitude data could be achieved. The fitting function $H$, based on the Nelder–Mead search principle, and the filter function $K$ are described in detail below.

After inputting the current function parameter $\boldsymbol{P}_\mathrm{m}$ and the current attitude data $X_{(n|n)}$, the fitting function $H$ substituted parameter $\boldsymbol{P}_\mathrm{m}$ into the residual calculation function $R$. If the optimization result did not meet the judgment threshold, parameter $\boldsymbol{P}_\mathrm{m}$ was substituted into the Nelder–Mead search algorithm $f_\mathrm{nm}$ for continuous optimization until the optimal solution was found, and the optimized function parameter $\boldsymbol{P}_{\mathrm{m}+1}$ was obtained. The specific process is shown in Equation (2).

$$H\left(\boldsymbol{P}_\mathrm{m}, X_{(n|n)}\right) \begin{cases} \boldsymbol{X}_{(n|n)}^\mathrm{nm} = \begin{bmatrix} X_{(n-l_\mathrm{en}|n-l_\mathrm{en})} & \cdots\cdots & X_{(n-1|n-1)} & X_{(n|n)} \end{bmatrix} \\ R_\mathrm{es} = R\left(\boldsymbol{P}_\mathrm{nm}, \boldsymbol{X}_{(n|n)}^\mathrm{nm}\right) = \dfrac{\sum\limits_{k=0}^{l_\mathrm{en}} \left(F(\boldsymbol{P}_\mathrm{nm}, t_\mathrm{n}) - X_{(n-k|n-k)}\right)^2}{l_\mathrm{en}} \\ \boldsymbol{P}_\mathrm{nm} = E_{xpand}(\boldsymbol{P}_\mathrm{m}) \\ \boldsymbol{P}_\mathrm{min} = f_\mathrm{nm}(R, \boldsymbol{P}_\mathrm{m}) \end{cases} \tag{2}$$

In Equation (2), the function $f_\mathrm{nm}$ comes from the NMSS (Nelder–Mead simplex search method) in [39], taking $\boldsymbol{P}_\mathrm{m}$ as the starting point to find a better solution to satisfy the minimum value of the function $R$, denoted as $\boldsymbol{P}_\mathrm{min}$. $\boldsymbol{X}_{(n|n)}^\mathrm{nm}$ is a satellite attitude data group which is consisting of saved attitude data. The length of the saved attitude data is $l_\mathrm{en}$ and the last data set is $X_{(n|n)}$. $R$ is an optimized judgment function based on the attitude data set $\boldsymbol{X}_{(n|n)}^\mathrm{nm}$ and the attitude prediction function $F$. The variance is represented by $R_\mathrm{es}$, which was used to judge the matching degree of the attitude data and the prediction function. $\boldsymbol{P}_\mathrm{nm}$ is the parameter expansion data group generated by the continuous cycle of reflection, expansion, contraction, and retraction shrink (marked as $E_{xpand}(\boldsymbol{P}_\mathrm{m})$) for parameter $\boldsymbol{P}_\mathrm{m}$ in NMSS. By continuously substituting the optimized parameter $\boldsymbol{P}_\mathrm{m}$ into $f_\mathrm{nm}$ and judging the

threshold value of $R_{es}$ until it met the fitting matching conditions, the optimized parameter $\boldsymbol{P}_{min}$ was used as the output $\boldsymbol{P}_{m+1}$ of the function $H$.

There is often a certain error in the data obtained by attitude acquisition equipment. The directly obtained attitude data are marked $Data(t_n)$, where $t_n$ is the time mark at time n given by the synchronous clock. The Kalman-like filter is marked $K$. The process is shown in Equation (3).

$$X_{(n|n)} = K\left(X_{(n|n-1)}\right) = flag \times X_{(n|n-1)} + (1 - flag) \times Data(t_n)$$

$$flag = \begin{cases} 0 & \left|Data(t_n) - X_{(n|n-1)}\right| \leq T_{h\_match} \\ 0.5 & T_{h\_match} < \left|Data(t_n) - X_{(n|n-1)}\right| \leq T_{h\_valid} \\ 1 & \left|Data(t_n) - X_{(n|n-1)}\right| > T_{h\_valid} \end{cases}$$

$$(3)$$

Kalman filtering is a cycle of prior estimation, the construction of the error covariance matrix, the construction of the correction matrix, and the update of the observed values to achieve the purpose of filtering. In the existing extended Kalman filter algorithm, the Jacobian matrix is usually used to fit nonlinear data linearly. Since the attitude change in a satellite is affected by the space environment, such as flywheel vibration, electromagnetic interference, solar storm, etc., the attitude uncertainty will change with the occurrence of events, and the probability density distribution of the uncertainty cannot be determined. Therefore, the KLFF algorithm refers to the idea of a priori estimation and data correction in the Kalman filter, uses the satellite attitude change model to perform a priori estimation, and corrects the actual measured value. Different from the Kalman filter, in the KLFF algorithm, the attitude data obtained by the high-precision attitude measurement equipment are relatively precise values, so there is no uncertainty characterization of the data. In Equation (3), $X_{(n|n-1)}$ is the estimated attitude at time n calculated by the fitting function $F$ in Equation (1), and the Kalman filtering of the prior estimated and collected attitudes is realized simply by judging the marker $flag$, in which the prior estimated attitude $X_{(n|n-1)}$ and the collected pose $Data(t_n)$ are calculated and compared with the threshold. $T_{h\_match}$ is the attitude mismatch threshold set according to the pixel mismatch and does not exceed 1/3 of the pixel or 1/25 of the attitude amplitude. $T_{h\_valid}$ is the abnormal attitude value threshold set to 1/5 of the attitude amplitude in this algorithm. By introducing the attitude function $F$, the collected data $Data(t_n)$ were substituted into the Kalman-like filter $K$, and finally the filtered attitude data $X_{(n|n)}$ for fitting the function $H$ were obtained.

The corresponding parameter prefitting module and adaptive real-time fitting module were designed based on the KLFF algorithm, and the process is shown in Figure 3. In the prefitting module, the initial parameters were calculated by performing Fourier transformation on the attitude data, and then the optimal fitting result of the fitting function $H$ was obtained using the NMSS. The method obtained is often a local optimal value, so the idea of an annealing algorithm can be introduced to perform suboptimal fitting through different amounts of data. Judging the fitting result and the optimal fitting threshold and finally obtaining the global optimal result, we constructed the attitude function. The process of the adaptive fitting module was as follows: First we judged whether the attitude measurement value was valid through the confidence interval. If valid, it was given a synchronization time stamp and used as the input data set $Data(t_n)$. Then, $Data(t_n)$ was substituted into the KLFF algorithm, and the adaptive optimization of the attitude function parameter $\boldsymbol{P}_m$ was realized through the cooperative calculation of the fitting function $H$ and the Kalman-like filter function $K$. Unlike the conventional extended Kalman filter principle, the collected attitude data were not extended by nonlinear data every time, but a satellite attitude change model was introduced. The data were first judged for outliers and then compared with the estimated value of the attitude function. By making an a priori judgment about the data, saving computing power and realizing real-time self-adaptation of the algorithm were achieved.

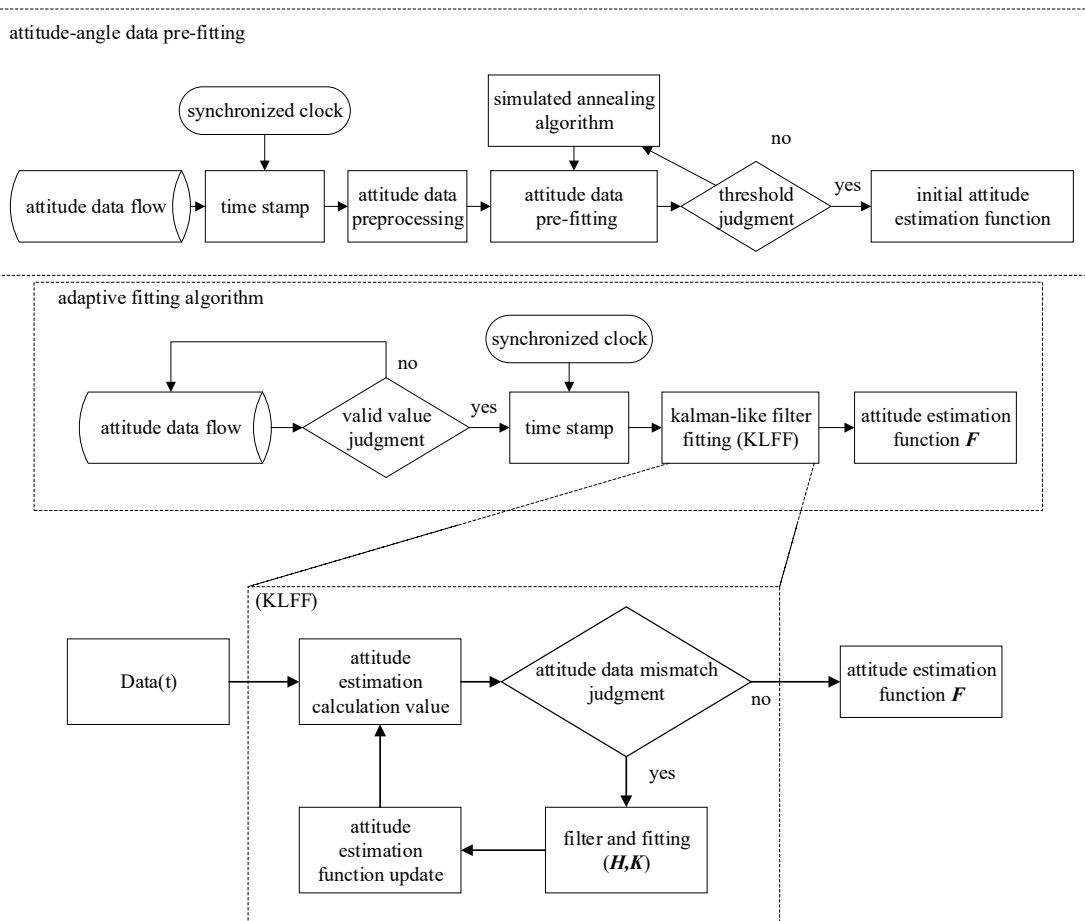

**Figure 3.** Adaptive attitude function fitting flow chart of Kalman-like filter fitting (KLFF).

### 2.2. Image Pixel Offset Calculation Principle

After the attitude estimation function was obtained through the adaptive fitting module, its relationship with the optical axis of the optical system needed to be constructed. It can be seen from Equation (1) that the fitted attitude estimation function expression is $F(\boldsymbol{P}_{\mathrm{m}}, t_{\mathrm{n}}) = \varepsilon + A_{\mathrm{mp}} \times \sin(\omega \times t_{\mathrm{n}} + \varphi)$, where the function parameter $\boldsymbol{P}_{\mathrm{m}}$ is determined, and the satellite attitude can be directly calculated by the estimation function. The satellite attitude is represented by a matrix of Euler angles. Since there was an installation angle error between the satellite attitude and the imaging coordinate systems, it was necessary to rotate the former to obtain the latter. According to the subsecond installation accuracy in [18] and the method of coordinate system conversion in [40], the fixed transformation matrix $\boldsymbol{M}$ was constructed. The attitude and the optical axis change in the imaging system were correspondingly solved through the internal or external rotation of the coordinate system, and the optical axis of the optical system was obtained. The conversion diagram is shown in Figure 4, and the conversion relationship is shown in Equation (4).

$$
\begin{cases}
\boldsymbol{\theta}_{\mathrm{st}} = \begin{bmatrix} X_{\mathrm{pitch}} & X_{\mathrm{yaw}} & X_{\mathrm{roll}} \end{bmatrix}^{\mathrm{T}} \\
\boldsymbol{\theta}_{\mathrm{opt}} = \boldsymbol{M} \times \boldsymbol{\theta}_{\mathrm{st}} \\
\boldsymbol{M} = \begin{bmatrix}
\cos\alpha\cos\beta & \cos\alpha\sin\beta\sin\gamma - \sin\alpha\cos\beta & \cos\alpha\sin\beta\sin\gamma + \sin\alpha\cos\gamma \\
\sin\alpha\cos\beta & \sin\alpha\sin\beta\sin\gamma - \cos\alpha\cos\gamma & \sin\alpha\sin\beta\sin\gamma + \cos\alpha\cos\gamma \\
-\sin\beta & \cos\beta\sin\gamma & \cos\beta\cos\gamma
\end{bmatrix} \\
\boldsymbol{\theta}_{\mathrm{opt}} = \begin{bmatrix} \theta_{\mathrm{x}} & \theta_{\mathrm{y}} & \theta_{\mathrm{z}} \end{bmatrix}
\end{cases}
\tag{4}
$$

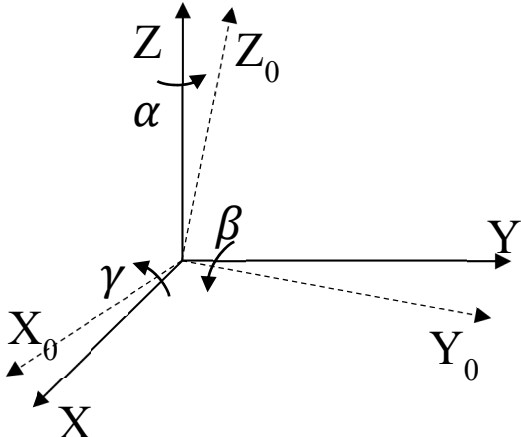

**Figure 4.** Schematic diagram of coordinate system conversion between attitude and imaging system.

In Equation (4), $\theta_{\text{st}}$ is the attitude estimation result calculated by the estimation function $F$ in Equation (1). $X_{\text{pitch}}$, $X_{\text{yaw}}$, and $X_{\text{roll}}$ represent the satellite attitude change values estimated by the estimation function $F$ on their respective axes. $M$ is the fixed transformation matrix of the satellite attitude and the imaging system obtained by rotating the coordinate system, wherein $[\alpha \; \beta \; \gamma]$ are the rotation angles around the $z$-, $y$-, and $x$-axes, respectively. $\theta_{\text{opt}}$ is the estimation result of the converted optical axis deflection angle, $[\theta_{\text{x}} \quad \theta_{\text{y}} \quad \theta_{\text{z}}]$ representing the optical offset angle with respect to the initial optical axis position of each coordinate system axis.

Taking the time stamp of the image as the input quantity $t$ and substituting it into the transformed optical axis change function $\theta_{\text{opt}}$, the offset of the image was geometrically solved through the optical structure to obtain the image pixel offset. The process is shown in Figure 5, the geometrical optics solution model is shown in Figure 6, and the calculation relationship is shown in Equation (5).

$$\begin{cases} L_{\text{pixel}_n} = f \times \tan(\theta_{\text{x}}) \\ \Delta L_{\text{nm}} = L_{\text{pixel}_n} - L_{\text{pixel}_m} \\ P_{\text{nm}} = \frac{\Delta L_{\text{nm}}}{S_{\text{pixel}}} \\ P_{\text{c}_{\text{move}}} = P_{\text{c}_n} - P_{\text{c}_m} \\ \Delta pixel = |P_{\text{nm}} - P_{\text{c}_{\text{move}}}| \end{cases} \quad (5)$$

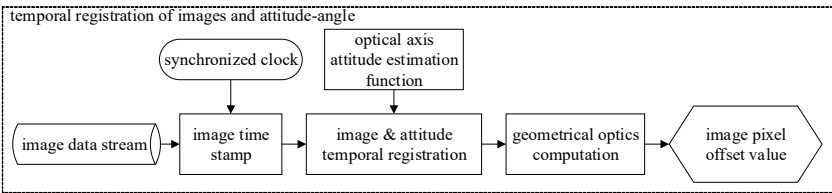

**Figure 5.** Image and attitude temporal registration flow chart.

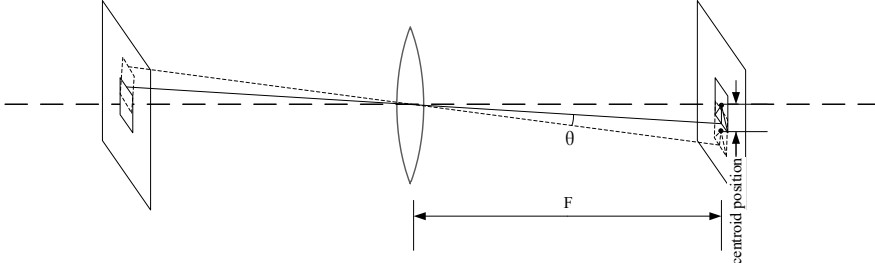

**Figure 6.** Schematic diagram of angular offset conversion of optical structure.

In Equation (5), $\theta_x$ is the deflection angle of the optical axis of the imaging system in the $x$-axis, and the same is true for other axes; $f$ is the optical focal length of the system; $L_{\text{pixel}_n}$ is the image displacement corresponding to the angle at time n; $\Delta L_{\text{nm}}$ is the distance offset at different times; $S_{\text{pixel}}$ is the pixel size of the imaging system; $P_{\text{nm}}$ is the image pixel offset calculated by the angle of the optical axis; $P_{\text{cn}}$ is the position of the image centroid at time n; $P_{\text{c}_{\text{move}}}$ is the pixel offset of the centroid at different times; and $\Delta pixel$ is the error of the algorithm calculation offset and the centroid extraction offset to evaluate the accuracy of the attitude estimation algorithm.

### 2.3. Simulation Experiment Verification and Data Acquisition

The simulation experiments were conducted by ocean observation or using laboratory equipment. In the ocean observation experiment, an infrared camera produced by the France Sofradir captured an image of the ocean surface, and SURF feature matching was used to extract and match the imaging features. The laboratory experiment was set to verify the algorithm. The simulated experimental environment was based on the satellite parameters. In the experiment, we used the H-824 precise 6-axis rotating platform produced by the German Physik-Instrumente company (Karlsruhe, Germany) to simulate the attitude change in the satellite, the gyroscope was fixed to the detection camera using the HWT6053 angle sensor of the WIT-MOTION company (Shenzhen, China), and the static measurement accuracy of the pitch angle and roll angle was 0.001°. The dynamic measurement accuracy was calibrated by using a turntable. At a low frequency of 0.003 Hz, the pitch angle and roll angle measurement accuracy was 0.002°, and the maximum data collection frequency was 25 Hz; that is, the fastest time was 40 ms to obtain attitude data. The camera was a Hikvision MV-CA060 (Hangzhou, China) with a resolution of 3070 × 2048 pixel and a pixel size of 2.4 μm. We used the camera's pixel combination mode (binning) to increase the signal-to-noise ratio, so the actual pixel size was 4.8 μm. The camera was equipped with a fixed-focus lens with a focal length of 50 mm. Its structure and the experimental equipment are shown in Figure 7.

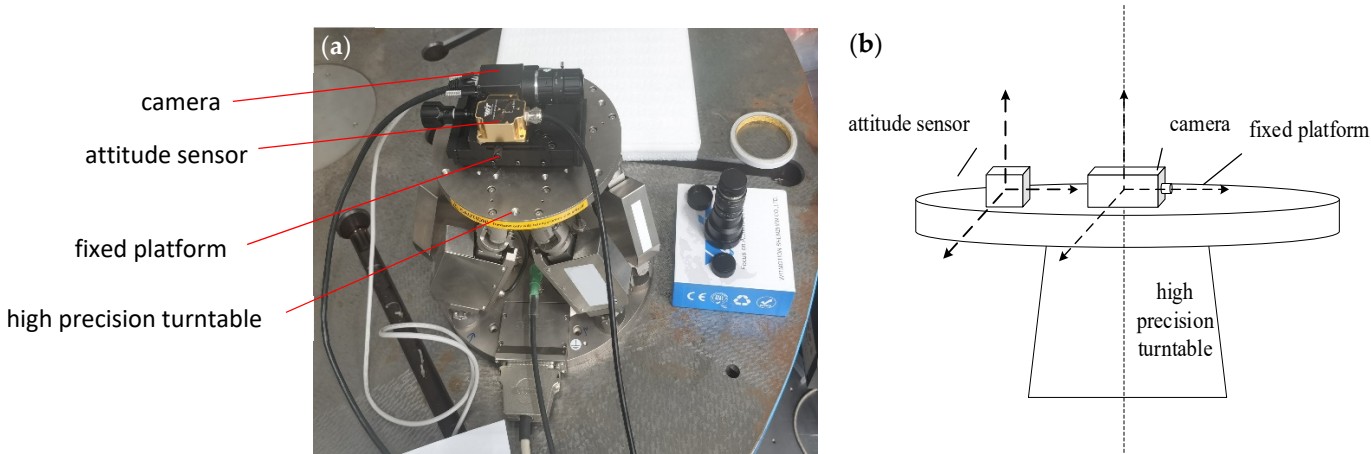

**Figure 7.** Experimental equipment and simulation experiment structure. (**a**) Experimental equipment; (**b**) simulated experimental structure.

Referring to the satellite parameters in [41], the image acquisition frame rate was 50 Hz. When the satellite attitude was not under image stabilization control, the low-frequency, large-angle amplitude peak value was about 0.03°; the frequency was 0.298 Hz; and the angle change per unit sampling time was about $2 \times 10^{-4}$°. After image stabilization control, the peak amplitude of subtle disturbance was about $1.4 \times 10^{-3}$°; the frame rate of high-precision attitude return data was about 15 Hz; and the measurement accuracy was $1.5 \times 10^{-4}$°. Considering the equipment accuracy of the simulation experiment, the

sampling frame frequency of the camera and the high-precision attitude information were proportionally reduced to simulate the satellite parameters, which are shown in Table 1.

**Table 1.** Comparison of the actual parameters of the satellite and the design parameters of the simulation experiment.

| Parameter Type | Frame Interval (ms) | Attitude Frequency (Hz) | Data Ratio | Attitude Amplitude (°) | Microvibration Amplitude ($10^{-3}°$) | Attitude Change Rate ($10^{-3}°$/s) | Sampling Accuracy ($10^{-3}°$) |
|---|---|---|---|---|---|---|---|
| Actual parameters of the satellite | 20.0 | 0.300 | 1:3 | 0.03 | 1.4 | 0.2 | 0.15 |
| Design parameters of the simulation experiment | 3000.0 | 0.003 | 1:2 | 0.05 | 5.0 | 4.0 | 2.00 |

## 3. Results and Discussion

### 3.1. Ocean Observation Simulation Experiments

In the ocean observation simulation experiments, multiple groups of infrared cameras were used to image and stitch the ocean. The effect after SURF feature extraction is shown in Figures 8 and 9.

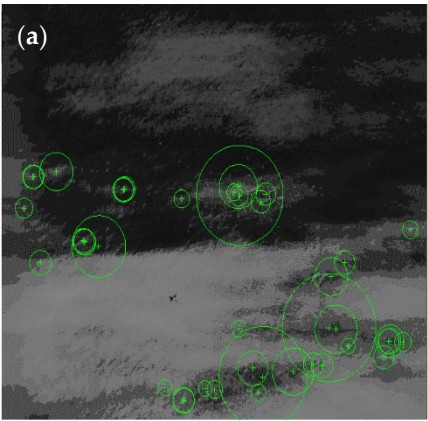 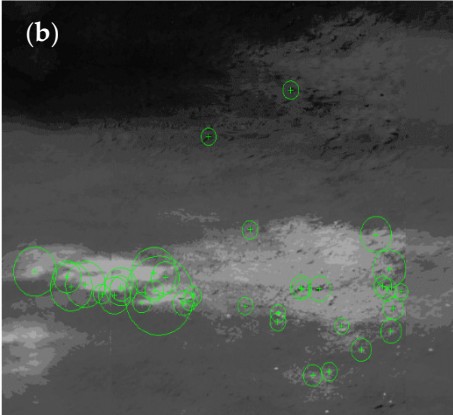

**Figure 8.** Extraction of feature points from infrared images of the ocean surface. (**a**,**b**) are the distributions of image feature points at different times.

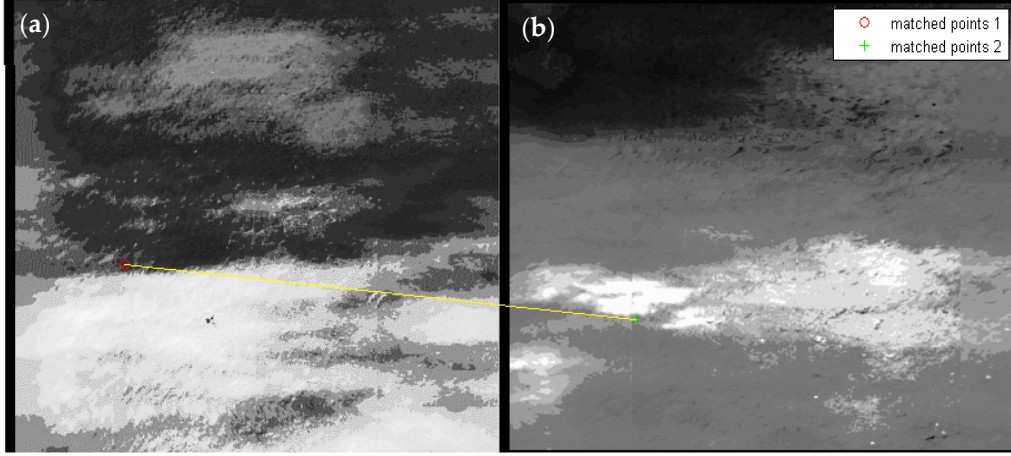

**Figure 9.** The error result of ocean image feature matching. (**a**,**b**) are the distributions of image feature points at different times.

In Figure 8, the feature points extracted by the SURF algorithm are relatively random because the characteristic objects on the ocean surface are not obvious. In Figure 9, there is an incorrect matching case, indicating that when factors such as waves or surface reflections change, the feature points also change. The resulting large matching errors make it difficult to confirm the image pixel offset. Therefore, in a high-orbit ocean imaging system, the commonly used feature-matching algorithm has certain practical limitations, so it is necessary to predict and fuse the satellite attitude data with the image formed by the imaging system.

### 3.2. Laboratory Equipment Simulation Experiments

To verify the attitude estimation algorithm, equipment simulation experiments were carried out in a laboratory. To evaluate the image pixel offset, its target used an image with obvious edge features to facilitate the calculation of the accurate pixel offset, which was used as the evaluation standard of the prediction algorithm, as shown in Figures 10 and 11. In the laboratory equipment simulation experiment, the frame rate of the image was about twice the attitude acquisition frequency. When the measurements of the satellite attitude were used to register the image directly, the image pixel offset estimation was incorrect due to the frequency difference. Figure 10 shows the result of image displacement judgment by edge extraction after subtracting the experimental target image and its two adjacent images. Figure 11 is a data representation diagram. The centroid extraction algorithm was used to solve the initial position of the image to obtain the precise offset angle of the subsequent image. The attitude data were high-precision and measured by the angle sensor.

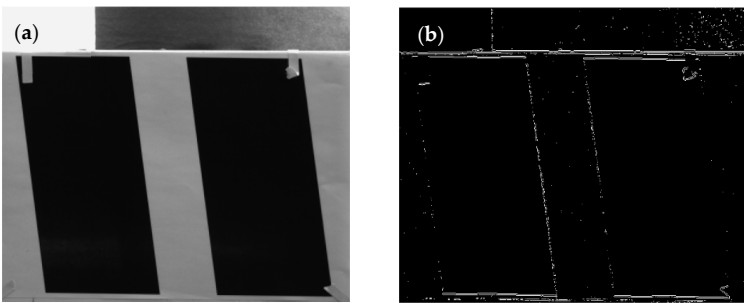

**Figure 10.** Experimental image processing results. (**a**) Experimental target image; (**b**) the result of subtracting edge extraction from adjacent frame images.

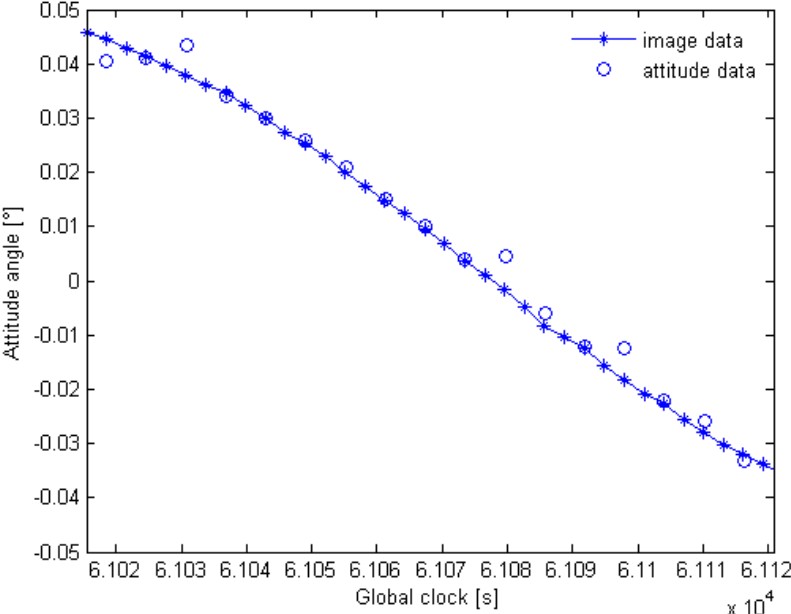

**Figure 11.** Distribution map of image acquisition data and attitude acquisition data.

As can be seen in Figure 10, the two adjacent frames of images had a certain pixel offset. Referring to Figure 11, when the attitude data could not be temporally registered with the image data, the actual pixel offset of two adjacent image frames could not be represented by discrete attitude data. Without function fitting, it was impossible to predict the image pixel offset with high accuracy, and if the image registration criterion was directly performed, edge blurring, as shown in Figure 10b, appeared. Therefore, it was necessary to use the KLFF algorithm to register the image with the attitude data to realize its instantaneous attitude estimation. For the collected discrete attitude data, the algorithm was used to fit the discrete attitude data and compare them with the attitude interpolation algorithm. The results are shown in Figure 12.

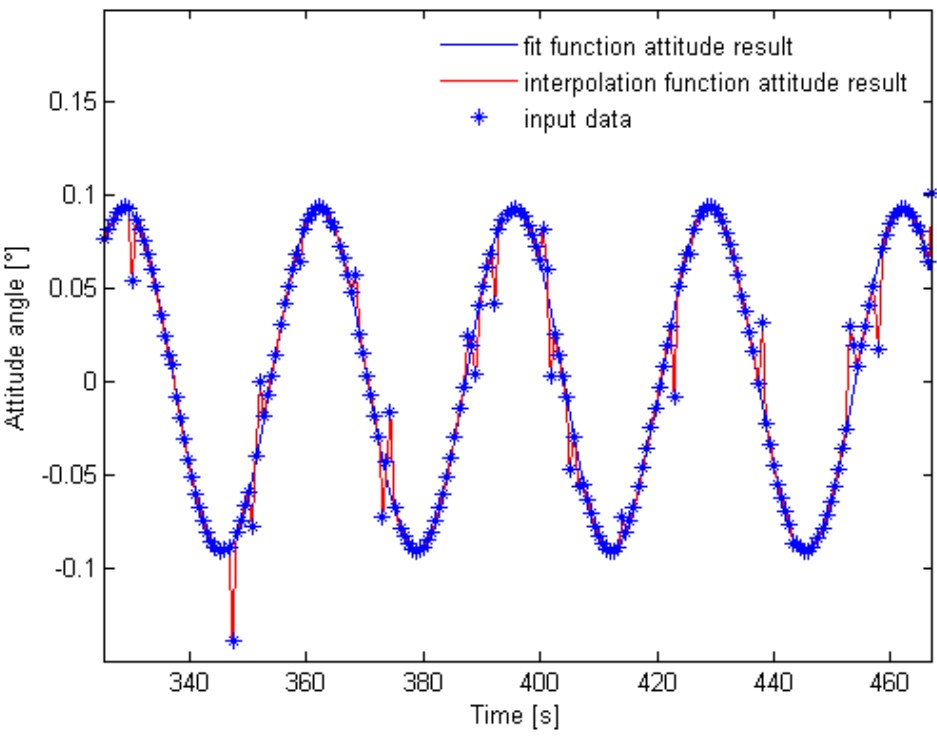

**Figure 12.** Optical axis angle fitting results of different fitting algorithms.

It can be seen from Figure 12 that the fitting algorithms described to a certain extent the change law of the attitude data. However, when there was jitter noise in the attitude measurement, the error was not filtered because the interpolation algorithm had sampled and interpolated the adjacent data. Furthermore, in a real-time attitude acquisition system, the latest attitude data are usually located at the edge of the stored data group. Traditional filtering techniques such as median filter, smoothing filter, etc., can perform better noise filtering on the data when the data are located in the middle of the data group, but the filtering effect is not good for edge data. If the data group is filtered after the attitude data are stored, the image data cannot obtain the real-time satellite attitude, resulting in the accumulation of image data. Therefore, this method introduced noise into the interpolation result, resulting in a large error in the offset angle of the optical axis. The Kalman-like filter algorithm, after obtaining information for each attitude, compared the measured attitude with the estimated value calculated by the attitude estimation function and realized real-time filtering of singular values. When the noise was random, the fitting process again performed a noise-averaging process on the attitude signal. Therefore, the algorithm fit discrete data better and had better noise filtering, compared with commonly used interpolation methods, which verified the feasibility of the attitude-fitting prediction algorithm.

Combined with the abovementioned adaptive-fitting algorithm, the image pixel offset was calculated. The pixel size of the imaging camera was 4.8 μm. Due to the limited

measurement accuracy of the simulated experimental equipment, the installation accuracy of the optical system was much higher than the measurement accuracy. Therefore, the conversion-fixed matrix *M* was approximated as a constant value 1. The experimental system had been calibrated, and the actual focal length of the imaging system was 47.5 mm. The image extracted by the centroid is shown in Figure 13. After the attitude fitting and geometric calculation, the image pixel offset estimation results are shown in Figures 14 and 15.

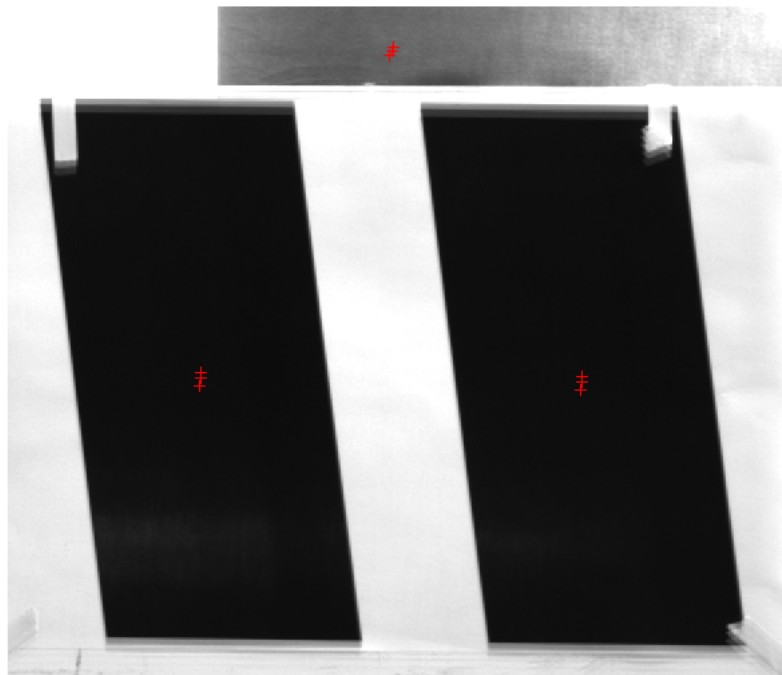

**Figure 13.** Extraction map of centroid at different moments when attitude changes.

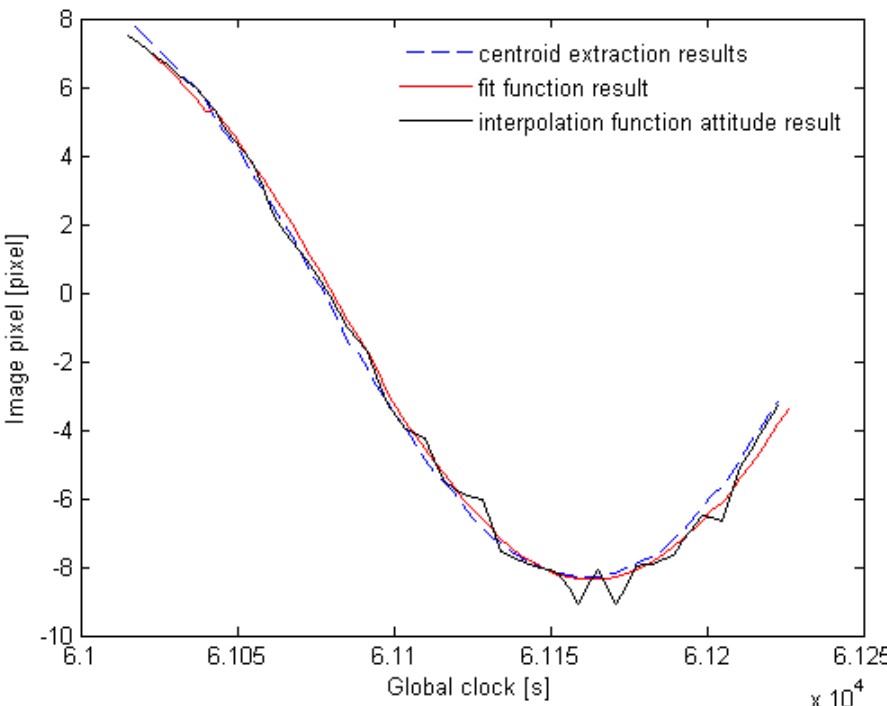

**Figure 14.** Image pixel offset estimation results for different fitting algorithms.

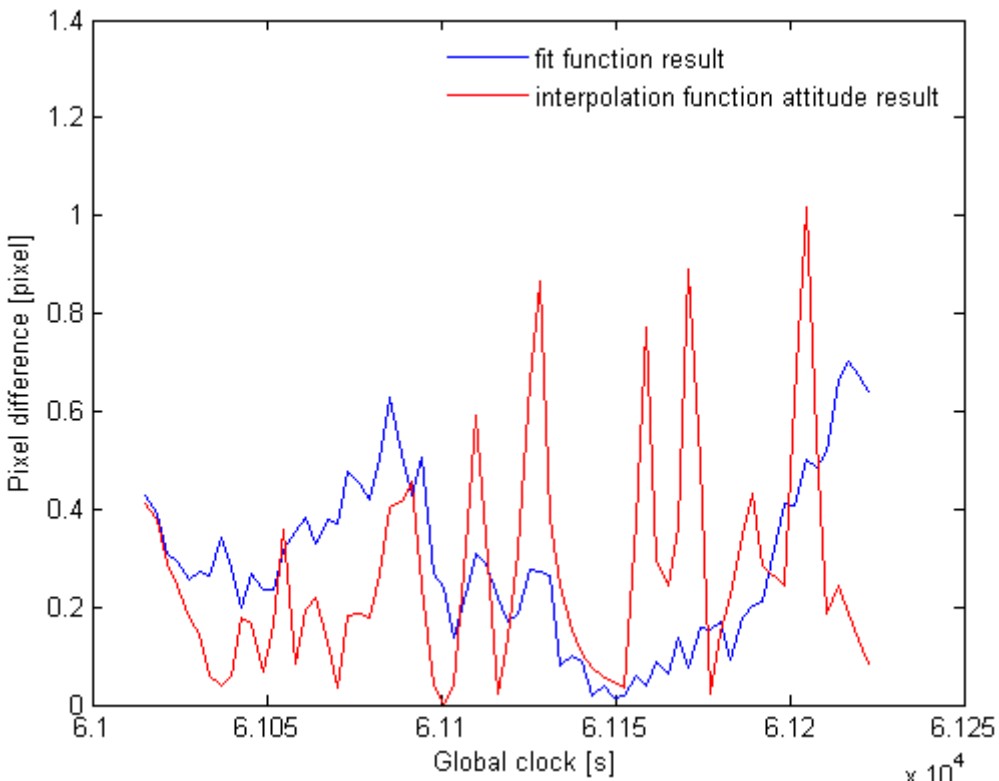

**Figure 15.** Image pixel estimation difference for different fitting algorithms.

Figure 13 shows that the centroid extraction accurately extracted the pixel offset distance of the image, so the instantaneous posture of the image obtained using the centroid extraction procedure was used as the ideal posture. As can be seen from Figure 14, after using the fitting algorithm by performing data fusion and registration on the high-precision time-marked image and discrete attitude data, the algorithm realized the calculation of the image pixel offset without using the feature point extraction algorithm, thereby realizing the offset registration between different images. At the same time, compared with the time-weighted interpolation method, the image pixel offset estimated by the algorithm had a higher degree of fit with the real offset. The error calculation results are shown in Table 2. Compared with the traditional interpolation attitude estimation algorithm, the estimation accuracy of the optical axis offset angle after using the algorithm was accurate from $6.4 \times 10^{-3\circ}$ to $3.5 \times 10^{-3\circ}$. Corresponding to a 4.8 μm pixel conversion, the image pixel offset value is shown in Figure 15: the mean value decreased from 0.292 to 0.165 pixels, and the peak value decreased from 1.112 to 0.602 pixels. It can be seen that after fusion of the attitude data, the image pixel offset was better estimated. Compared with traditional interpolation algorithms, the algorithm in this paper had better prediction accuracy and stability.

**Table 2.** Estimated error results after processing by different algorithms.

| Fitting Method | Angle Difference Mean $(10^{-3\circ})$ | Angle Difference Peak $(10^{-3\circ})$ | Pixel Difference Mean (Pixel) | Pixel Difference Peak (Pixel) |
|---|---|---|---|---|
| Time weight interpolation | 1.69 | 6.41 | 0.292 | 1.112 |
| Adaptive Kalman-like filter | 0.96 | 3.51 | 0.165 | 0.602 |

After the high-precision estimation of the pixel offset of the image, the linear interpolation method was used to realize the image pixel offset compensation on the vertical axis of the image (*y*-axis), since the imaging system had sinusoidal attitude changes on this

vertical axis. The horizontal axis (*x*-axis) represented the natural jitter error when there was no attitude change, which was used as a comparison item. At the same time, the results were compared with the traditional interpolation algorithm, and the results are shown in Figure 16.

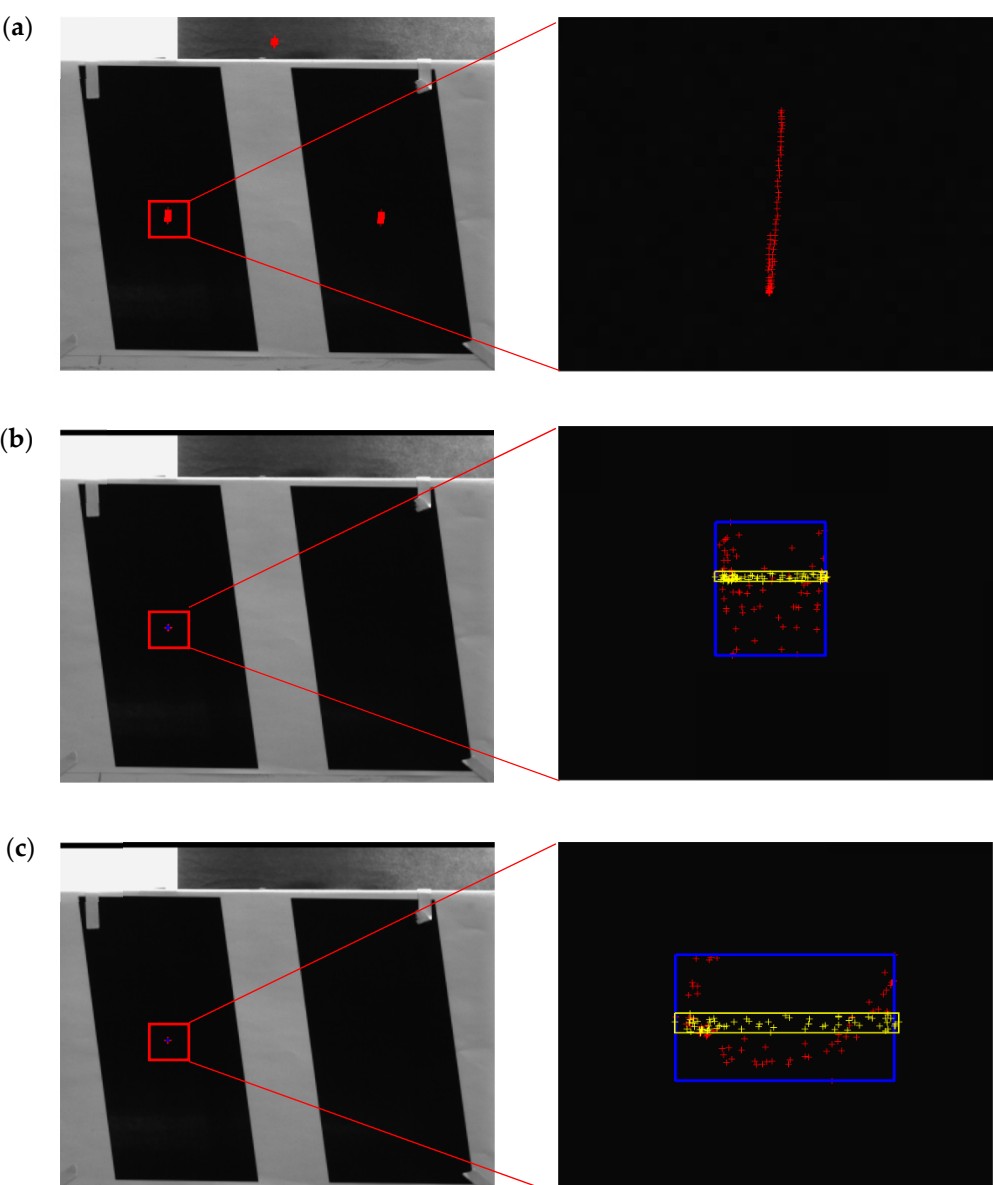

**Figure 16.** Distribution of centroids after image pixel offset compensation for different algorithms: (**a**) distribution of the centroid when the vertical axis has attitude changes; (**b**) distribution of the image pixel offset compensation centroid after using the attitude estimated by the traditional interpolation algorithm; (**c**) distribution of the image pixel offset compensation centroid after using the attitude estimated by the KLFF algorithm. The red dots are the image centroids of different frames, the blue matrix is the image pixel offset compensation centroid after using the KLFF algorithm, and the yellow matrix is the image pixel offset compensation centroid after centroid extraction for comparison.

The estimated attitude was calculated using the KLFF algorithm, and the vertical axis offset of the image was offset according to the estimated attitude. The centroid position of the compensated image in each frame is shown in Figure 16b. The rectangle represents the degree of centroid aggregation after image pixel offset compensation. The blue matrix

is the image pixel offset compensation centroid after using the KLFF algorithm, and the yellow matrix is the image pixel offset compensation centroid after centroid extraction for comparison. The more the centroid of the blue rectangle deviates from the centroid of the yellow matrix, the greater the error in attitude estimation accuracy; that is, the greater the estimation error. The larger the blue rectangle, the more discrete the image pixel offset compensation; that is, the worse the predicted noise resistance and algorithm stability. Compared to Figure 16a, the attitude estimated by the KLFF algorithm characterized the image pixel offset well. After compensation, the offset error of the algorithm was 0.029 pixels, and the image pixel offset compensation discrete range of the vertical $y$-axis was 0.697 pixels. Compared with the jitter range of 1.144 pixels on the horizontal $x$-axis without attitude correction, the dispersion was reduced by 39.07%. It can be seen from Figure 16c that due to the introduction of attitude measurement noise, the image pixel offset compensation effect of the traditional interpolation algorithm was not ideal. The offset error was 0.137 pixels, and the $y$-axis image pixel offset compensation dispersion range was 1.476 pixels. Compared with the traditional interpolation algorithm, the offset error was reduced from 0.137 to 0.029 pixels after compensation and accuracy increased by 78.8%. At the subpixel level, it had an improved effect, but it was not obvious. The discrete range was reduced from 1.476 to 0.697 pixels (52.8%), and the effect of noise was reduced from the pixel level to the subpixel level. Compared with no attitude correction and traditional interpolation algorithms, the algorithm effectively achieved noise reduction in attitude data, thereby achieving the high-precision estimation of the instantaneous satellite attitude and image pixel offset.

### 3.3. Algorithm Stability and Real-Time Analysis

After adding random noise to the experimental attitude data, the experiment was repeated 100 times, and the repair effect of the algorithm was compared with that of the interpolation algorithm. The results are shown in Figure 17.

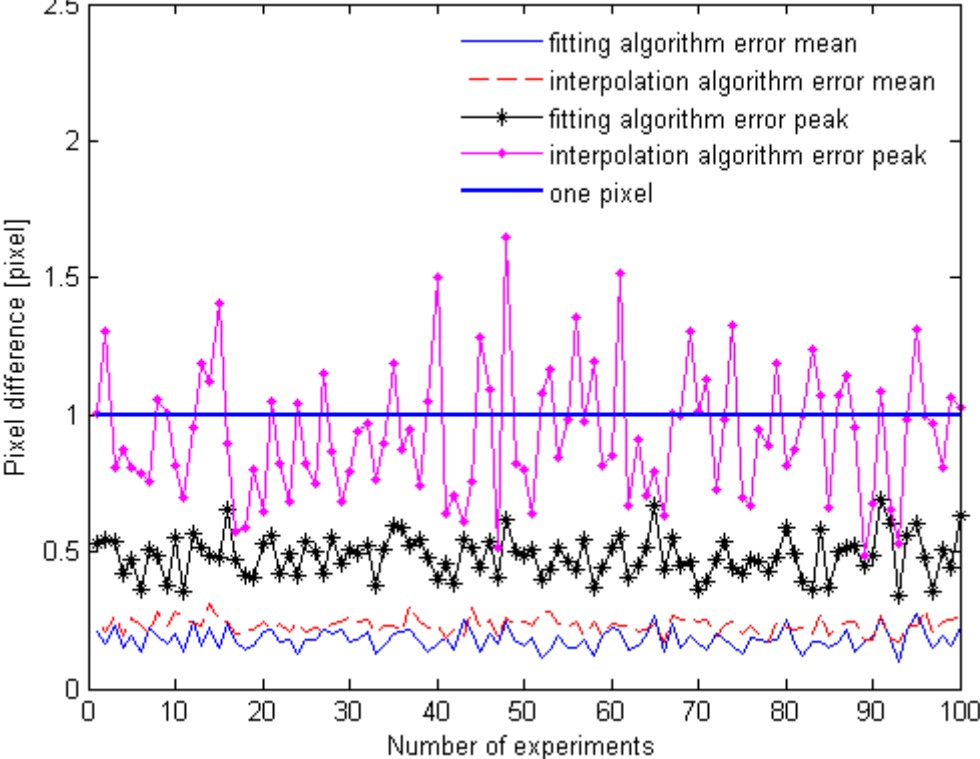

**Figure 17.** Pixel estimation error results after repeated experiments with different algorithms.

It can be seen from Figure 17 that compared with the time-weighted interpolation method, the algorithm was optimized for fitting error and algorithm noise resistance, and the results are shown in Table 3. The average range of the optical axis angle error fitted by the algorithm was from $0.58 \times 10^{-3\circ}$ to $1.6 \times 10^{-3\circ}$, and the peak range was from $1.9^\circ \times 10^{-3\circ}$ to $4.0^\circ \times 10^{-3\circ}$. The time-weighted interpolation method was used in the comparison group, and the average value of the obtained optical axis angle error ranged from $0.98 \times 10^{-3\circ}$ to $1.8 \times 10^{-3\circ}$, and the peak range was from $2.8 \times 10^{-3\circ}$ to $9.6 \times 10^{-3\circ}$. When the image pixel size was 4.8 μm after the geometrical optics calculation, the optical axis angle offset corresponding to 1 pixel was $5.7 \times 10^{-3\circ}$. The corresponding pixel offset value is shown in Table 3.

**Table 3.** Estimated error results of different algorithms after repeated experiments.

| Fitting Method | Fit Error Mean Range | | Fit Error Mean Median | | Fit Error Peak Range | | Fit Error Peak Median | |
|---|---|---|---|---|---|---|---|---|
| | Angle $(10^{-3\circ})$ | Pixel (Pixel) | Angle $(10^{-3\circ})$ | Pixel (Pixel) | Angle $(10^{-3\circ})$ | Pixel (Pixel) | Angle $(10^{-3\circ})$ | Pixel (Pixel) |
| Interpolation | 0.98 ~1.8 | 0.169 ~0.310 | 1.39 | 0.2395 | 2.8 ~9.6 | 0.484 ~1.658 | 6.2 | 1.071 |
| KLFF | 0.58 ~1.6 | 0.100 0.276 | 1.09 | 0.1880 | 1.9 ~4.0 | 0.328 ~0.691 | 2.5 | 0.509 |
| Lift Ratio (%) | / | / | 21.58 | 21.50 | / | / | 59.68 | 52.43 |

In Table 3, the mean prediction error represents the prediction accuracy of the KLFF algorithm, and the peak value of the prediction error represents the prediction stability. Under repeated experiments, compared with the time-weighted interpolation algorithm, the KLFF algorithm reduced the mean median prediction error by about 0.1 pixels, and the prediction accuracy showed an average improvement of 21.50%. The median prediction error peak value decreased from 1.071 to 0.509 pixels, from pixel to subpixel level, and the prediction stability and robustness greatly improved for an average improvement of 52.43%. Referring to the requirement that the pixel mismatch of the satellite system was less than 1/3 pixel, after introducing the external measurement attitude, fitting, and predicting, the fitting results characterized the real image pixel offset better. Compared with the traditional interpolation algorithm, the estimation error of the algorithm was accurate from the pixel to the subpixel level. This showed that the KLFF algorithm was less affected by attitude measurement noise from the attitude sensors. This improved the stability and the noise immunity and anti-interference ability of the algorithm while ensuring a high-precision estimated attitude.

To verify the real-time performance of the KLFF algorithm, a simulation experiment was carried out on the time consumption image pixel offset calculation and attitude function fitting. In the experimental system of this manuscript, the CPU used was AMD 5800X, the reference frequency was 3.80 GHz, and the GPU was not used for computing acceleration. Moreover, we used the method of image interpolation to explore the time consumption of image pixel offset compensation. The results are shown in Figure 18.

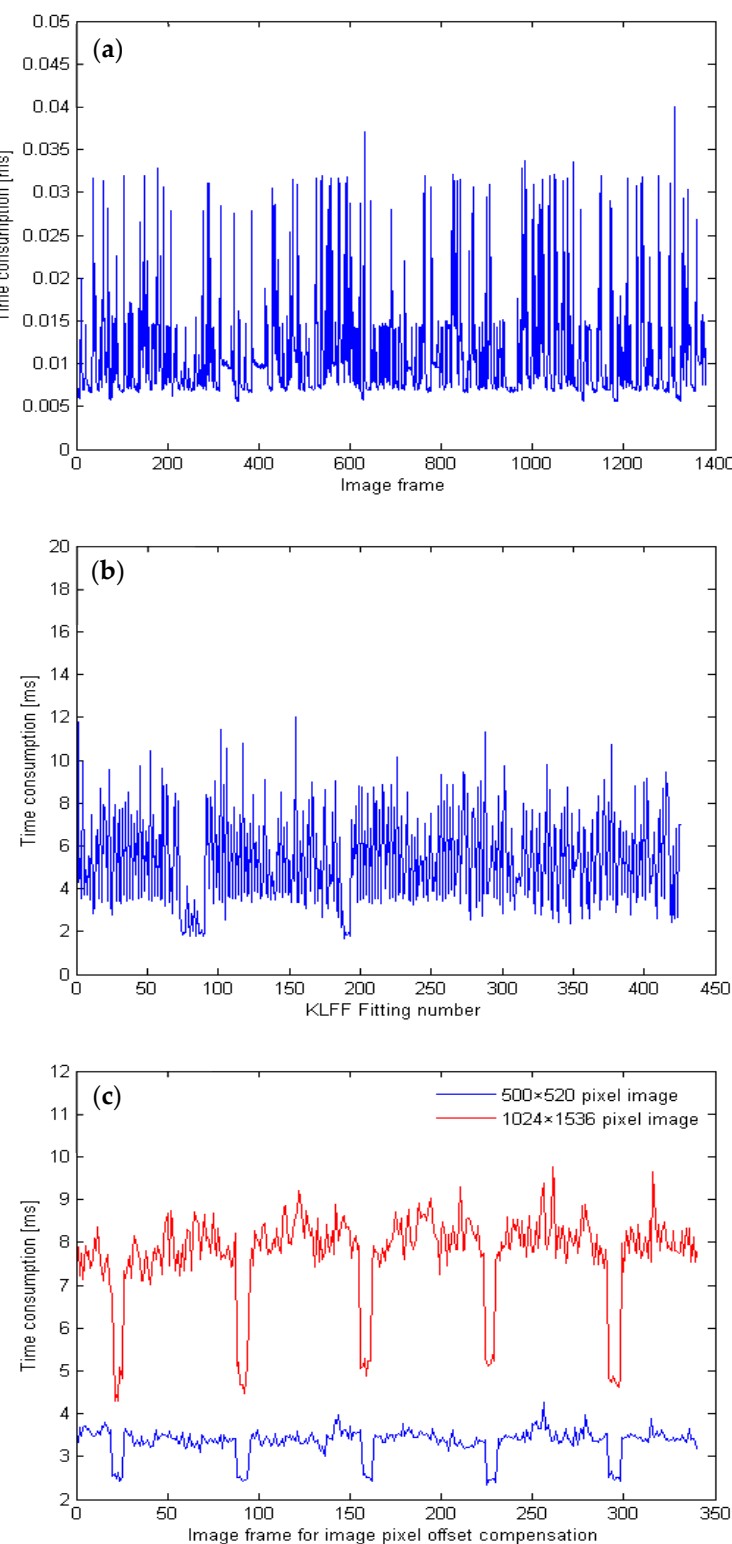

**Figure 18.** Algorithm time consumption: (**a**) time consumption of each frame of image pixel offset calculation; (**b**) time consumption situation of each KLFF algorithm fitting; (**c**) time consumption situation of image pixel offset compensation for each frame under different windowing modes.

The algorithm in this paper decoupled the fitting of the attitude estimation function from the calculation of the image pixel offset. Compared with commonly used feature extraction methods, it did not perform feature processing on the image first, but it directly realized the solution of the image pixel offset by substituting the time stamp of the image

for the pose estimation function. The computing power it required involved a simple function calculation. Figure 18a shows the time consumption for each image frame to calculate the image pixel offset. Peak consumption was 0.04 ms. Compared with the 100 Hz (10 ms) frame rate of the camera, with the help of the attitude prediction function, the simple attitude calculation met the real-time requirements of high-frame-rate images. Figure 18b shows the time consumption of each attitude function fitting. The first fitting process corresponded to the prefitting module. Since the initial input parameters were relatively rough, it took a long time: 18 ms. After applying adaptive fitting, the peak time consumption of attitude fitting was 12 ms compared with the 15–20 Hz (50–66.7 ms) frequency of high-precision attitude measurement equipment. As can be seen in Figure 18c, when the image pixel offset shows a sinusoidal variation law, the time consumption of image processing has a law similar to that of a square wave. The least time-consuming part of image processing corresponds to the peak and trough positions of the image pixel offset, where the pixel offset is small, so the pixel offset compensation takes less time. The image pixel offset in other positions is larger, so the image processing time consumption will also increase accordingly. In this experiment, the image windowing mode used was $500 \times 520$ pixels. The image pixel data were less, so the speed of pixel offset compensation was faster. The peak processing time of each frame was 4.25 ms, which could meet the camera's limit frame rate of 100 Hz (10 ms). As a comparison, when the windowing mode is $1024 \times 1536$ pixels, due to the increase in processing data, the peak processing time of pixel offset processing is 9.5 ms. At this time, the time consumption of pixel offset compensation is relatively close to the limit frame rate of 100 Hz (10 ms), so there is a risk of pixel compensation overtime. Therefore, when the image pixel offset calculation and the attitude estimation function fitting run in parallel, the algorithm can meet real-time requirements to a certain extent. In addition, when we perform offset compensation for image pixels, the real-time performance of image processing will change according to the computing power of the processing equipment and the difference in the amount of image data. In the experiment in this paper, the pixel offset compensation took little time and could meet the camera's limit frame rate of 100 Hz (10 ms). However, for other time-consuming situations, certain methods need to be adopted to increase the image processing speed, such as reducing the limit frame rate of the camera, dividing and parallel-processing the image, etc. After using the KLFF algorithm in this manuscript to obtain the image pixel offset, how to increase the speed of image processing is an important and independent research direction, and a lot of research needs to be carried out in the future.

## 4. Conclusions

To obtain better image quality and a higher signal-to-noise ratio, a geostationary orbiting ocean imaging system needs to maintain higher platform stability over a longer integration time. To reduce this requirement, pixel offset compensation is often performed on the image, and this requires high precision. For marine targets, due to the lack of image features, it is difficult for the image pixel offset compensation system to use a conventional feature-matching algorithm to calculate the image pixel offset. Therefore, in this paper we introduced external measurement data, fused the attitude data with the image, estimated the satellite attitude, and calculated the image pixel offset without using feature matching. A Kalman-like filter fitting (KLFF) algorithm based on a satellite attitude change model was designed. First, a synchronous clock was used to give time stamps to the images and satellite attitudes; then, Kalman-like filtering was performed on the satellite attitude data, and real-time adaptive function fitting was performed on the satellite attitude change model based on the Nelder–Mead search method. Finally, the fitted attitude estimation function and synchronization time stamp were used to estimate the satellite attitude and image pixel offset with high precision.

The average error between the image pixel offset estimated by the KLFF algorithm and the actual offset was 0.1880 pixels, which realized the high-precision prediction of the image pixel offset. After using the estimated attitude of the KLFF algorithm for image pixel

offset compensation, a comparison with the traditional interpolation algorithm revealed that the offset error was reduced from 0.137 to 0.029 pixels, and accuracy improved by 78.8%. The dispersion range of the image pixel offset compensation was reduced from 1.476 to 0.697 pixels, the dispersion was reduced by 52.8%, and the estimation error caused by noise was also reduced. The performance of the algorithm was verified by repeated experiments. Compared with the traditional interpolation method, the algorithm optimized the error peak value from 1.6 to 0.691 pixels, a reduction of about 1 pixel. The improvement was about 52.43%, indicating that the influence of noise on attitude estimation had been suppressed, which verified the algorithm's noise resistance and stability.

Furthermore, the equipment parameters and algorithm time consumption analysis verified real-time performance. Without using feature extraction, the high-precision estimation of the instantaneous attitude of the satellite and image pixel offset was realized based on the satellite attitude change model. The accuracy and noise immunity of the attitude estimation algorithm improved, which laid the foundation for the high-precision registration of image pixel offsets. Moreover, this method avoids feature loss and computing power consumption when using feature extraction, which improves the real-time performance and lays the foundation for on-orbit implementation.

However, the algorithm has high synchronization requirements for the system clock and requires a time reference with high precision and small drift. It is also necessary to know the attitude model of the satellite operation in advance. Only after the satellite attitude change model was constructed could the algorithm be used for attitude estimation, which suggested research requirements for constructing a satellite attitude model. Because of the randomness of measurement noise, the algorithm can reproduce the optimization effect of attitude estimation but cannot 100% reproduce the data in this manuscript. In the future, in order to solve the universality of the algorithm, the satellite attitude change law should be modeled first. After a satellite attitude change model is constructed, the algorithm can be used for attitude estimation. It is also possible to further improve the accuracy of the algorithm by studying a high-precision, high-stability time synchronization system. In addition, some research on high-speed image processing is also required. We need to accelerate the image processing speed to ensure that we can use high-precision image pixel offsets to perform image pixel offset compensation in real time.

**Author Contributions:** Conceptualization, L.H., F.D. and Y.F.; Data curation, L.H.; Formal analysis, L.H.; Investigation, L.H.; Methodology, L.H. and Y.F.; Project administration, F.D. and Y.F.; Resources, F.D. and Y.F.; Supervision, F.D. and Y.F.; Validation, L.H.; Writing—original draft, L.H.; Writing—review and editing, F.D. and Y.F. All authors have read and agreed to the published version of the manuscript.

**Funding:** This research was funded by Research on the Mechanism and Model of Ocean Optical Remote Sensing Detection, grant number 2016YFC1400901.

**Data Availability Statement:** The data presented in this study are available on request from the first author.

**Acknowledgments:** The authors would like to thank Key Laboratory of Infrared System Detection and Imaging Technology, Chinese Academy of Science, for supporting this research work. This project has been supported by Research on the Mechanism and Model of Ocean Optical Remote Sensing Detection.

**Conflicts of Interest:** The authors declare no conflict of interest. The funders had no role in the design of the study; in the collection, analyses, or interpretation of data; in the writing of the manuscript; or in the decision to publish the results.

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
