# Peer review of "Prediction Algorithm for Satellite Instantaneous Attitude and Image Pixel Offset Based on Synchronous Clocks"

_remotesensing, doi:10.3390/rs14163941_

Round 1

Reviewer 1 Report

The authors have devoted great efforts to revise the manuscript. The manuscript has been significant improved. I recommend the revised version.

Author Response

We are very grateful to the expert for your guidance and advice on this manuscript, and we express our highest gratitude and respect to you.

Reviewer 2 Report

The manuscript is a revised version of the manuscript “Research on Instantaneous Attitude Prediction Algorithm of Satellite Imaging Based on Time Synchronization System” by the same authors. The authors took into consideration my comments to the original version of the manuscript. As a result, the manuscript has improved. I have, however, the following specific comments. After taking these comments into consideration, I guess, the manuscript can be accepted for publication.

Specific comments

Line 3: The words “synchronous time” are nowhere used in the manuscript, except in the title. It is not clear, what synchronous time is. Therefore, I suggest to use the words “on Synchronous Clocks” instead of “on Synchronous Time” in the title.

Lines 16 and 528: I think, you should use the words “satellite attitude change” instead of “satellite motion” to avoid misunderstanding. Otherwise, it sounds like you mean satellite motion around the Earth. Please, check also other cases, when the word “motion” is used in the manuscript. I think, more proper use would be “attitude change” rather then “motion”, since this is what you mean.

Line 58: the words “the range 10–20Hz” should read “the range of 10–20Hz”.

Lines 124-125. The following sentence should be reformulated “Based on the characteristics of the image and satellite attitude data, we constructed the corresponding experimental and data model, as shown in Figure 1.” No model is shown in Figure 1, but just two schematic diagrams.

Line 154: the words “specific time at time stamp” should better read as “specific time instant at time stamp”.

Line 174-175: “is a satellite attitude data group”. Do you mean “being a satellite attitude data group”.

Line 175: the words “length of saved attitude data” should better read “length of the saved attitude data”.

Line 220: The title of subsection 2.2 should better read “Image pixel offset calculation principle”.

Lines 240-241: the words “the converted optical axis estimation result” are not clear and should be reformulated. Which result?

Lines 241-242: “the optical offset angle of each coordinate system axis.” Offset angle with respect to what?

Line 276: it is better to write the complete word “pixel” instead of “px”, as it is done in other cases in the manuscript.

Line 278: “um”. If it stands for micrometer, then it is better to write it in the same way, as done on line 276.

Table 1. I think it is better to write the words “Satellite parameters” and “Simulation experimental parameters” in a consistent way to the words “actual parameters of the satellite and the design parameters” of the table capture, otherwise it is not clear, what is what.

Line 299-300: “The effect after SURF feature extraction was performed”. Do you mean “The effect after SURF feature extraction was observed”.

Line 313: please explain in the manuscript, what you mean under the words “attitude of the image”. It is used also later, but is not explained.

Line 324: “its instantaneous attitude”. Do you mean here attitude of the satellite or image? Please, specify in the manuscript.

Line 389: the words “data fusion of the attitude data” should better read “fusion of the attitude data”.

Line 396: the words “to offset the vertical y-axis” are not clear. Please, reformulate.

Line 449. Symbols “3.” just under the Table 3 should be erased.

Line 491: “offset compensation”. Please, specify: offset between what?

Line 519: “anti-noise”. What is that? I am not sure that such a word exist.

Line 524: “requirements of the system clock” should better read as “requirements for the system clock”.

Author Response

Thanks to the expert for the comments on this manuscript, I have benefited greatly from your suggestions. With reference to these suggestions, we have conducted a series of discussions and analyses, and the details are in the attachment .

Reviewer 3 Report

The paper “Research on Instantaneous Attitude Prediction Algorithm of Satellite Imaging Based on Time Synchronization System” investigates the problem of satellite attitude estimation. It proposes to use data from the conventional satellite sensors and simultaneously utilize images to get additional information and improve the attitude determination accuracy. Although this topic is interesting, I cannot recommend this paper for publication due to the following reasons.

Critical issues

1)      The authors propose Kalman filtering technique for attitude determination. However, they do not provide the satellite model of motion and model of measurements. They just provide vague description of the algorithm. The utilized model of motion is simple periodic functions, and this is justified by referring to a master thesis which I couldn’t even find at the provided link.

2)      The mathematics used in equations 1,2 looks incorrect.

3)      The image processing usually requires a lot of computational time. I could not find any discussion on how 100 Hz rate can be ensured in real-time attitude estimation. In addition, authors tell in the paper (lines 309-316) that the feature extraction algorithm faces difficulties when works with oceanic surface. Therefore, I doubt that this algorithm can be implemented on-board the satellite.

4)      The paper still requires extensive proof reading.

Author Response

(The authors gave the same response as above.)

Round 2

Reviewer 3 Report

The paper “Prediction Algorithm for Satellite Instantaneous Attitude and 2 Image Pixel Offset Based on Synchronous Clocks” suggests a technique of attitude measurements filtering using “Kalman-like filter fitting”. As I mentioned in previous review, this topic is of great interest, although, in my opinion, this work has some critical flaws to be published in such a highly rated journal as Remote Sensing

Critical issues

1)      Kalman filters take into account motion model and measurement model. In addition, they allow us to take into account measurements noise, and provide data that allow us to estimate the filter convergence (i.e. the covariance matrix). Suggested in the paper approach (although it utilizes the simple sinusoidal model of motion) does not take into account the measurements noise, and is actually just a least square method.

2)      In my opinion, comparison of the suggested technique with conventional interpolation method is incorrect. The authors did not apply any conventional filtering techniques to the raw data before passing it to the interpolation function. I believe it would greatly affect the final results.

3)      The mathematics in Eq.4 is incorrect. The final angles are not defined by provided equations.

4)      Authors should provide information about the system that conducted the calculations (at least CPU characteristics). Without that, the time consumption of the suggested technique cannot be verified.

Author Response

Dear expert:

Thanks to the expert for the comments on this manuscript, I have benefited greatly from your suggestions. With reference to these suggestions, we have conducted a series of discussions and analyses, and hereby give some feedback to you. Please see the attachment for details.

This manuscript is a resubmission of an earlier submission. The following is a list of the peer review reports and author responses from that submission.

Round 1

Reviewer 1 Report

Unfortunately, I cannot recommend this paper for publication. See the attached PDF for more details

Reviewer 2 Report

The manuscript describes the algorithm on instantaneous attitude prediction of geostationary satellite imaging based on using time synchronization system developed by the authors and provides some results of application of this algorithm. Some parts of the manuscript should be improved to make them more clear according to the comments given below. After taking these comments into consideration, a decision on manuscript acceptance for publication can be made.

A general concept comment

The algorithm developed by the authors should be described in more details. For example, it should be explained what the attitude of the image (lines 67 and 402) is? How is it related to the satellite attitude? It should be specified, how attitude is measured (Line 115). It should be clearly formulated, what is new in proposed algorithm as compared to those used before. Is the algorithm developed by the authors reproducible? Is the software of the developed algorithm freely available? Can the attitude prediction algorithms developed by the authors be applied to any satellite, not necessary geostationary one? I think, it is important to address these questions.

Specific comments

Line 11: “image volume of the imaging image” should better read “volume of the image” or even “image volume”

Lines 10-15, 19-22, 22-27, 129-131, 133-137, 137-141, 146-150, 216-221, 221-227, 227-230, 230-233, 238-242, 263-268, 305-308, 309-313, 334-341, 382-386, 391-398, 406-417, 417-421 and some other cases: these sentences are too long. Please split them into few sentences. This can be easily done in most cases by replacing symbols “;” by “.”

Line 17 and other parts of the manuscript: I suggest to replace “imaging image” by “image”.

Lines 22, 24, and 25: the degree of 10 should be written in the upper row, in which symbol ° is located.

Lines 26-27: “The estimated stability”. Stability of what? Please, specify.

Line 37: a space is missing between “image” and “[1]”.

Line 40: please, explain the following abbreviations at their first use: SIFT, SURF.

Line 48 and other parts of the manuscript: please put a space between a value and its unit, e.g. “100Hz” should be written as “100 Hz” and so on.

Line 50: 3.19x10-4°: the degree -4 of 10 should be written in the upper row.

Line 76: It is important to mention that precise attitude modeling is important also for such satellites as altimetry (Bloßfeld et al., 2020) and GNSS satellites (Loyer et al., 2021).

Line 83: “measurement angle”. Please, specify the angle between which directions.

Line 93: “the stability of the whole star” should better read “the stability of the whole star image”.

Line 103: “the angle and attitude changes”. Which angle? Please, specify.

Lines 107, 110 and 111: “data model” and “System Model”. What is shown in Figure 1 are, from my point of view, not models, but just illustrations.

Line 116: “global synchronization clock”. What does this clock represent?

Line 137: a space is missing inside “Data(t).Substitute”.

Line 155: I wonder, if formulae can be provided for H and F in Equation (1).

Lines 177 and 178: “in reference [18]” should read “in [18]” and “in reference [33]” should read “in [33]”.

Line 182: “is shown in 4” should read “is shown in Fig. 4”.

Line 182 and similar cases in Lines 188 and 198: “is shown in formula 2” should read “is shown in Eq. (2)”.

Line 188: “estimated attitude estimation result” should better read “estimated attitude”.

Lines 190-191: the text “[α β γ] is the rotation angle around the z-axis, the rotation angle around the y-axis and the rotation angle around the x-axis” can be better written as “[α β γ] are the rotation angles around the z-axis, y-axis and x-axis, respectively”.

Line 233: a point is missing at the end of the sentence.

Line 248: the first word of each column header should start with a capital letter, namely, “frame” should be written as “Frame”.

Lines 251 and 272: the subsection titles should start with a capital letter.

Line 252: “infrared imaging cameras” should be better written as “infrared cameras”.

Line 254: “is shown in Figure 7”. In fact, there is no Figure 7. There are Figures 7.1 and 7.2, instead. I would suggest a consistent figure numbering, like Fig. 7, 8, 9, etc.

Line 264: should “Surf” be written in capital letters as “SURF” for consistency with line 40?

Line 265: the words “incorrect matching case in the matching” should be, from my point of view, reformulated. I guess “incorrect matching case” is enough.

Line 277: “shown in Figure 8”. In fact, there is no Figure 8. There are Figures 8.1 and 8.2 instead.

Line 279: “acquisition attitude”. What is acquisition attitude?

Fig. 8.2, Fig. 9, Fig. 10.1, Fig. 10.2, Fig. 11: the units in the X and Y axes should be written as [°] and [s] [pixel] instead of /°, /s and /pixel, respectively.

Lines 323 and 332: “shown in Figure 9”. In fact, there are two Figures 9 in the manuscript, but should be just one. Please, check figure numbers.

Line 325: “10.1 and 10.2.” A word “Fig.” should be added before the figure numbers.

Line 332: “centroid extraction” should better read as “centroid extraction procedure”.

Line 334: a low case symbol ° before the words “As can be seen from Figure 10.1” should be erased.

Line 345-346 and 378: the word “increased” should be replaced by the word “decreased” (twice), since the values decreased.

Table 2: the first word of each column header should start with a capital letter, namely, “pixel” should two written as “Pixel”.

Lines 359-360: English in this sentence should be improved. Presently, it sound as “comparison… are both optimized… ”.

Lines 361 and 367: “shown in Table 3”. In fact, there is no Table 3. There are Tables 3.1 and 3.2 instead. Please, better use continues counting of tables and figures.

Line 386: “estimation algorithm”. Estimation of what? Please, specify.

Line 387: “the attitude error”. What is the attitude error for the satellite of your study? Please, specify.

Lines 405-406: “The error is less than 1/3 pixel, indicating that the algorithm in this paper can predict the instantaneous attitude of the image.” Please, explain why.

Line 408 and: “pixel accuracy” should read “pixel”, since the error can not be improved by an accuracy, but by a specified value.

Line 413: the word “improves the accuracy and accuracy” should be reformulated.

Lines 442-514: please provide missing doi numbers or web links to the references, if no doi number is available. This significantly simplifies access to the references.

Lines 480 and 514: only the first letters of the words of journal titles should be normally written in capital, all other letters are usually written in small letters.

References used in the review:

Bloßfeld, M.; Zeitlhöfler, J.; Rudenko, S.; Dettmering, D. Observation-Based Attitude Realization for Accurate Jason Satellite Orbits and Its Impact on Geodetic and Altimetry Results. Remote Sensing. 2020, 12, 682, https://doi.org/10.3390/rs12040682.

Loyer, S.; Banville, S.; Geng, J.; Strasser, S. Exchanging satellite attitude quaternions for improved GNSS data processing consistency. Advances in Space Research, 2021, 68 (6), 2441-2452, https://doi.org/10.1016/j.asr.2021.04.049.

Reviewer 3 Report

Overall, the manuscript is not well written. There are numerous issues in the writing style, which requires intensive revision to make the manuscript to be followed.

In this manuscript, the authors attempt to  obtain the satellite attitude using equipment such as gyroscopes, and perform time registration between the attitude and the imaging image data, and achieve offset matching between images. The idea is fine, but not be well presented the motivation and scientific new contributions.

I provide the following comments for the authors to consider:

(1) The writing of this manuscript does not satisfy the standard of an journal article yet. There are numerous issues or mistake. For example, Line 12, "time; Due" is an obvious mistake. Please check the whole manuscript if they would like to revise the manuscript.

(2) The section "Introduction" is the weakest part of the manuscript. The first paragraph is too long, which can be 2 paragraphs. The authors need to highlight the importance of the scientific problem considered in this manuscript. At the end of the section, it is necessary to clearly highlight or identify the limitation of the existing works and research gaps of the current knowledge, new scientific contribution and motivation of this manuscript; and the aims of this study.

(2) I don't think it is necessary to have figure number or table as Figure 3.1, 3.2, Table 3.1, 3.2, etc. If they are one figure of one table, you should only have one figure or table number.

(3) Figure caption, table caption, section headings, sub-section headings, the first chapter should be caption.

(4) In the section "Method", it is unclear which  parts are the authors' new contributions. It seems all come form previous references or textbook. The authors may need to highlight their new contribution.

(5) In the section "Results and discussion", the authors need to provide more clear structure and in-depth discussion.

(6) "Conclusions" is unclear. The authors need to intensive revise and summarise the finding in a better way.

(7) References need to be consistent in the format. For example, References 1, 7 & 18. There either format issue of no volume number (xx) is certainly incorrect.